# Complement 1q/Tumor Necrosis Factor-Related Proteins (CTRPs): Structure, Receptors and Signaling

**DOI:** 10.3390/biomedicines11020559

**Published:** 2023-02-14

**Authors:** Constanze Schanbacher, Heike M. Hermanns, Kristina Lorenz, Harald Wajant, Isabell Lang

**Affiliations:** 1Institute of Pharmacology and Toxicology, University of Würzburg, Versbacher Str. 9, 97078 Würzburg, Germany; 2Department of Internal Medicine II, Division of Hepatology, University Hospital Würzburg, Auvera Haus, Grombühlstrasse 12, 97080 Würzburg, Germany; 3Leibniz-Institut für Analytische Wissenschaften—ISAS e.V., Bunsen-Kirchhoff-Str. 11, 44139 Dortmund, Germany; 4Department of Internal Medicine II, Division of Molecular Internal Medicine, University Hospital Würzburg, Auvera Haus, Grombühlstrasse 12, 97080 Würzburg, Germany

**Keywords:** adiponectin, AMPK, C1q/TNF related protein (CTRP), inflammation, metabolism

## Abstract

Adiponectin and the other 15 members of the complement 1q (C1q)/tumor necrosis factor (TNF)-related protein (CTRP) family are secreted proteins composed of an N-terminal variable domain followed by a stalk region and a characteristic C-terminal trimerizing globular C1q (gC1q) domain originally identified in the subunits of the complement protein C1q. We performed a basic PubMed literature search for articles mentioning the various CTRPs or their receptors in the abstract or title. In this narrative review, we briefly summarize the biology of CTRPs and focus then on the structure, receptors and major signaling pathways of CTRPs. Analyses of CTRP knockout mice and CTRP transgenic mice gave overwhelming evidence for the relevance of the anti-inflammatory and insulin-sensitizing effects of CTRPs in autoimmune diseases, obesity, atherosclerosis and cardiac dysfunction. CTRPs form homo- and heterotypic trimers and oligomers which can have different activities. The receptors of some CTRPs are unknown and some receptors are redundantly targeted by several CTRPs. The way in which CTRPs activate their receptors to trigger downstream signaling pathways is largely unknown. CTRPs and their receptors are considered as promising therapeutic targets but their translational usage is still hampered by the limited knowledge of CTRP redundancy and CTRP signal transduction.

## 1. Introduction

The coordinated communication of different organs is of fundamental importance for the maintenance of a healthy metabolic state. This task crucially depends on the activity of secreted endocrine hormones and cytokines. A conserved family of such secreted plasma proteins are the complement 1q (C1q)/tumor necrosis factor (TNF)-related proteins (CTRPs). The best and most intensively studied prototypical representative of this ligand family is adiponectin, an important regulator of the lipid and glucose metabolism. As summarized in a number of review articles, adiponectin increases insulin sensitivity of skeletal muscle, adipose tissue and liver. This results in enhanced glucose uptake into adipose tissue and utilization in skeletal muscle, decreased gluconeogenesis in the liver, enhanced lipid clearance and lipid uptake into adipocytes coupled to promotion of fatty acid oxidation [1,2,3,4]. Accordingly, the reduced production of adiponectin promotes the development of the metabolic syndrome. Indeed, clinical studies showed that low serum adiponectin concentrations correlate with type 2 diabetes, coronary artery disease and obesity. Besides adiponectin, 15 other structurally related ligands exist, from CTRP1 to CTRP15 (Table 1). A PubMed title/abstract search end of December 2022 for adiponectin and its receptors AdipoR1 and AdipoR2 retrieved more than 23,000 entries including many, often specialized reviews. In contrast, a corresponding search for the other CTRPs (including aliases) retrieved only approximately 800 entries. In this review, we focused on these latter, less investigated CTRP family members.

The CTRPs are typically expressed and secreted by fat cells, but can also be released by muscle and endothelial cells (Table 1). The analysis of CTRP knockout and CTRP transgenic mice along with in vivo overexpression studies with viral vectors suggests that the CTRPs could have similar functions as adiponectin in the regulation of sugar and lipid metabolism (Table 2). Serum levels of various CTRPs are associated with diseases such as diabetes and coronary heart disease. For example, CTRP9 abundance in the serum is reduced in diabetic mice or after ischemia/reperfusion (I/R) injury or increased in response to chronic pressure overload; CTRP3 levels are decreased in patients with coronary artery disease (CAD) whereas CTRP1 levels are increased in CAD and correlate with blood pressure and blood lipids [31,45,46,47,48]. It is thus of particular scientific and possible socio-economic interest to clarify whether recombinant CTRPs are suitable for the therapy of prevalent cardiovascular diseases or the currently world-wide increasing non-alcoholic fatty liver disease (NAFLD).

The heart is not only an important target organ of CTRPs, but it also produces several members of the CTRP family at high levels, in particular, adiponectin, CTRP1, CTRP4, CTRP7 and primarily CTRP9 which has the highest expression level in this organ [49,50]. Similar to other CTRPs that have by now been analyzed for their impact on the heart, CTRP9 revealed to be largely protective in different types of disease models including myocardial infarction, I/R injury, LPS-induced cardiac dysfunction, pulmonary arterial banding and atrial fibrillation [31,32,49,51,52,53,54,55] (Table 2). Cardiac protection after myocardial infarction and/or experimental I/R injury was also reported for CTRP1, CTRP2, CTRP3, CTRP12 and CTRP15 [56,57,58,59,60,61]. Strikingly, however, CTRP9 was found to aggravate pressure overload-induced heart failure suggesting that under these special conditions CTRP9 either exceeds an optimal concentration, or that CTRP9 might be processed differently resulting in altered receptor or co-receptor usage [45]. It is worth mentioning that viral overexpression of CTRP3 or CTRP15 attenuated the hypertrophic response, fibrotic remodeling and left ventricular dysfunction induced by chronic pressure overload [55,62], suggesting that different CTRPs do not simply act redundantly in the heart.

**Table 2 biomedicines-11-00559-t002:** CTRPs: lessons from animal models.

CTRP	Model	Effect	Refs.
CTRP1	CTRP1-KO ^1^	Defective blood pressure homeostasis	[63]
	CTRP1-KO	Low-fat diet: insulin resistance and hepatic steatosis High-fat diet: reduced body weight, adiposity and hepatic steatosis	[64]
	CTRP1-tg ^2^	Reduced DOX-induced cardiac injury	[65]
	Adeno-CTRP1 CTRP1-KO	Inhibition of neointimal thickening	[66]
	Recombinant CTRP1 in non-human primates	Antiplatelet thrombotic activity	[67]
	CTRP1-KO	Increased myocardial infarct size, cardiomyocyte apoptosis, and inflammatory gene expression after I/R	[59]
	CTRP1-KO	Increase in kidney weight, glomerular hypertrophy, and elevated blood pressure in aged male	[68]
CTRP2	CTRP2-tg	Improved insulin tolerance	[69]
	CTRP2-KO	Up-regulation of lipolytic enzymes	[70]
	Injection of recombinant CTRP2	Reduced infarct size and improved blood flow after I/R	[58]
CTRP3	ApoE−/− ^3^	Reduced CTRP3 expression	[71]
	Overexpression In vivo knockdown	Overpression: exacerbated cardiac hypertrophy after pressure overload Knockdown: reduced cardiac hypertrophy after pressure overload	[72]
	CTRP3-tg	Attenuated hepatic triglyceride accumulation after chronic alcohol consumption	[73]
	CTRP3-tg in myocardium	Protection against sepsis-induced myocardial dysfunction	[74]
	Adenoviral CTRP3	Alleviated cardiac fibrosis	[75]
	CTRP3-KO	Enhanced collagen-induced arthritis	[76]
	CTRP3-tg	High-fat diet: reduced hepatic steatosis	[47]
CTRP4	CTRP4-tg	Inhibition of colitis	[77]
	CTRP4-KO	Partly increased pain sensitivity Reduced hippocampal-dependent associative learning in females	[78]
	CTRP4-KO	Increased sensitivity for sepsis	[79]
	Brain injection of recombinant CTRP4	Reduced food intake and body weight	[22]
CTRP5	Lentiviral CTRP5	Inhibition of adipose tissue browning	[80]
	CTRP5-KO	Improved insulin activity, attenuated diet-induced hepatic steatosis	[20]
	AAV ^4^-CTRP5	Reduced cardiomyocyte apoptosis and injury after I/R	[81]
CTRP6	CTRP6-tg in cardiomyocytes	Cardioprotection against DOX ^5^	[82]
	CTRP6-KO	Improved insulin action	[25]
	Lentiviral CTRP6 shRNA	Protection against diet-induced obesity, enhanced adipogenesis	[83]
CTRP7	CTRP7-KO	Low-fat diet: indistinguishable from WT mice High-fat diet: attenuation of insulin resistance, decreased liver fibrosis	[26]
CTRP9	Recombinant CTRP9 intranasally	Neuroprotection	[84]
	Lentiviral-CTRP9 in ApoE−/−	High-fat diet: reduced lesion size, less macrophages	[85]
	CTRP9-KO, CTRP9-tg	KO: exacerbates myocardial I/R injury tg: protects against myocardial I/R injury	[86]
	CTRP9-KO, CTRP9-tg	KO: protects against TAC ^6^ cardiac hypertrophy tg: promotes hypertrophic remodeling and dysfunction after TAC	[45]
CTRP12	CTRP12 +/− male	High-fat diet: impaired lipid clearance, greater steatosis	[87]
	CTRP12-KO, Adenoviral CTRP12	KO: increased neointimal thickening after vascular injury, enhanced inflammation Adenoviral: reduced neoinitmal thickening, reduced inflammation	[88]
	Lentiviral CTRP12	Reduced atherosclerosis in ApoE−/− mice	[89]
CTRP13	Infusion of recombinant CTRP13 in ApoE−/−	Reduced atherosclerotic lesions	[90]
CTRP15	CTRP15-KO, CTRP15-tg	KO: enhanced I/R myocardial infarct size, cardiac dysfunction, inflammation tg: reduced myocardial damage and inflammation after I/R	[57]
	CTRP15-KO	High-fat diet: reduced physical activity of male, elevated triglyceride, impaired lipid clearance	[91]

^1^ KO, knockout. ^2^ tg, transgene expression. ^3^ ApoE, apolipoprotein E. ^4^ AAV, adeno-associated virus. ^5^ DOX, doxorubicin. ^6^ TAC, transverse aortic constriction.

## 2. Structure of CTRP Family Members

With the exception of CTRP4, all CTRPs consist of a variable N-terminal domain with conserved cysteine residues, a collagen domain, and a C-terminal trimerization domain structurally related to the tumor necrosis factor (TNF) homology domain (THD) of ligands of the TNF superfamily (TNFSF) and the complement component C1q (ref. [92], Figure 1A). CTRP4 has an unusual domain architecture with a second C1q domain instead of the collagen domain and CTRP15 has a very short collagen domain which divides the N-terminal domain ([43,93], Figure 1A).

The CTRPs assemble into homo- or heteromeric molecules via their eponymous C1q domain. The collagen domains of three CTRP protomers form a helix and thus additionally stabilize the trimeric C1q domain of the CTRPs. CTRP trimers can also assemble to oligomers by forming disulfide bridges between the cysteines within the variable N-terminal domains (Table 3). Soluble CTRP variants that only contain the C1q domain result from proteolysis or can be obtained by genetic engineering and are referred to as globular CTRPs (gCTRPs). As has so-far been studied, gCTRPs, like the full-length CTRPs, can also interact with their receptors. The C1q domain is thus not only structurally related to the THD of TNFSF ligands, but similar to this domain, also mediates the binding to cellular receptors. Interestingly, there is evidence that trimeric gCTRPs and oligomerized full-length CTRPs can differ in their activity, a phenomenon well known from the TNFSF field [96]. For example, trimeric adiponectin was found to activate the AMP-activated protein kinase (AMPK) signaling pathway, whereas oligomerized adiponectin trimers do not activate this signaling pathway but stimulate the NFκB signaling pathway [97]. Furthermore, it has been reported for murine CTRP12, which can be cleaved between the variable and collagen/C1q domain by furin, that full-length CTRP12 activates the Akt/PKB signaling pathway in H4IIE hepatocytes and 3T3-L1 adipocytes, while the C1q-containing cleavage product preferentially stimulates the MAP kinases ERK1/2 and p38 [98]. Furthermore, only full-length CTRP12 enhanced insulin-stimulated glucose uptake in adipocytes. CTRP12 processing appears to be enhanced in mouse models of diet-induced obesity [98]. Unusually, the C1q-containing cleavage product forms dimers instead of trimers [98]. In the course of studies on the cardioprotective function of CTRP9, it was also shown that full-length CTRP9 is processed by incubation with heart tissue extracts to gCTRP9, which then more potently stimulates AMPK and Akt/PKB signaling [99].

In contrast to homotrimerization and homo-oligomerization, which have been described for all CTRP family members, heteromerization has only been described for a few of the 120 possible combinations of CTRPs. Adiponectin forms heteromers with CTRP9 and CTRP12, CTRP2 with CTRP7, CTRP8 with CTRP9, CTRP10 with CTRP13 and CTRP15 forms heteromers with CTRP2, CTRP5, CTRP10 and CTRP12 (Table 3). CTRP heteromerization has mainly been demonstrated and studied upon transient coexpression but has partly also been shown in vivo. Adiponectin and CTRP9 mutually coimmunoprecipitated each other from the serum of mice expressing these CTRPs transgenically [50]. Likewise, endogenous CTRP10 coimmunoprecipitated with virally expressed CTRP13 in mice [42]. So far investigated, the variable domains and their cysteines play no role for CTRP heteromerization, instead it was found that the globular C1q domain can be sufficient for heteromerization [50,100]. In line with these findings, differently tagged CTRPs only coimmunoprecipitated upon coexpression, but not after mixing separately produced proteins [11,100]. This suggests that CTRPs associate during their biosynthesis before secretion into stable complexes.

The crystal structures of the globular C1q domains of adiponectin, CTRP5, CTRP10, CTRP13 and CTRP14 have been solved and revealed trimeric symmetric assembly closely related to those found for ligands of the TNFSF [101,102,103,104]. Intriguingly, the structures of globular adiponectin, gCTRP13 and gCTRP14 contained Ca^2+^ bound to conserved binding sites along the interface of the three CTRP protomers and on the trimer surface, which seem to stabilize the trimeric C1q domain assembly [101,102]. Globular adiponectin was crystallized in Ca^2+^-bound and Ca^2+^-free form revealing a less compact structure of the latter [101]. It is worth mentioning that the Ca^2+^ binding sites are not preserved in CTRP5 [104].

Many questions concerning the differential activities of gCTRP trimers versus full-length CTRP oligomers are, however, completely open. For example, it is unclear whether the qualitatively different activities of globular and full-length CTRPs reflect different activity states of the same CTRP receptor(s) or the use of different CTRP receptors. Likewise, the impact of the oligomerization state of CTRPs on their stability, tissue penetrance and serum half-life are unclear. There is a similar situation with respect to heteromerization of CTRPs. It is unclear how CTRP heteromerization affects CTRP receptor activities, CTRP receptor usage and engagement of CTRP-induced signaling pathways.

**Table 3 biomedicines-11-00559-t003:** CTRPs: homo- and heteromerization.

CTRP	Homo- and Oligomerization	Heterotrimerization with	Refs.
Adiponectin	Trimers, hexamers, octadecameric complexes, HMW complexes	CTRP2, CTRP9	[29,50,97,105]
CTRP1	Trimers	CTRP6	[11]
CTRP2	Trimers	Adiponectin, CTRP7	[11]
CTRP3	Trimers, higher-order oligomeric complexes	-	[11]
CTRP4	Trimers, higher-order oligomeric complexes	-	[22]
CTRP5	Trimers, higher-order oligomeric complexes	-	[11]
CTRP6	Trimers, higher-order oligomeric complexes	CTRP1	[11]
CTRP7	Trimers	CTRP2	[11]
CTRP8	Trimers	-	[29]
CTRP9	Trimers, higher-order oligomeric complexes	Adiponectin	[29,50]
CTRP10	Trimers, higher-order oligomeric complexes	CTRP13	[11,100,102]
CTRP11	Trimers, higher-order oligomeric complexes	-	[38]
CTRP12	Dimers, trimers, oligomers	-	[39,98]
CTRP13	Trimers, octameric-like or higher-order oligomeric complexes	CTRP10	[100,102]
CTRP14	Trimers	-	[102]
CTRP15	Trimers, higher-order oligomeric complexes	CTRP2, CTRP12 and weaker with CTRP5, CTRP10	[43]

## 3. CTRP Receptors and Signaling

### 3.1. CTRP Receptors

Two structurally related receptors for adiponectin were identified by the expression cloning and have been named as adiponectin receptor 1 and 2 (AdipoR1, AdipoR2) [5]. Both receptors contain seven transmembrane domains (7TM) with an intracellular N-terminus and an extracellular C-terminus distinguishing these receptors from the huge superfamily of 7TM G-protein-coupled receptors (GPCRs) with an “N-terminus out” topology [106,107] (Figure 2). Instead, AdipoR1 and AdipoR2 are grouped together with nine other phylogenetically related receptors with seven or eight transmembrane domains and an “N-terminus in” topology in the PAQR (progestin and adipoQ receptors) protein family [107]. Adiponectin oligomers also interact with their C1q domains with T-cadherin, a glycosylphosphatidylinositol-anchored molecule [6,108]. Knockdown studies suggest that AdipoR1 acts also as receptor for CTRP7 and CTRP9 [15,31,51,53].

Notably, some CTRP receptors belong also to the 7TM GPCR superfamily. The closely related CTRPs CTRP10, CTRP11, CTRP12 and CTRP14 interact with the thrombospondin repeat domain of brain-specific angiogenesis inhibitor 3 (BAI3), a cell-adhesion class GPCR. Relaxin/insulin-like family peptide receptor 1 (RXFP1 receptor) has been identified as a receptor for CTRP8 in human glioblastoma driving tumor cell migration and chemotherapy resistance [27,28,35,109]. The CTRP13-BAI3 interaction has been implicated in the formation of a trans-synaptic adhesion complex by the ability of CTRP13 to interact with neuronal pentraxin 1 (NPTX1) which in turn strongly binds the neuronal pentraxin receptor NPTXR [42].

Adiponectin and CTRP9 signal also the by help of receptors not belonging to the PAQR protein family. Adiponectin interacts with T-cadherin and CTRP9 furthermore binds N-cadherin and the Toll-like receptor-4 (TLR4)-MD2 complex [6,33]. Similar to AdipoR1 and AdipoR2, T-cadherin was identified as an adiponectin receptor by expression cloning [6]. Of note, it was found that only hexameric and oligomeric but not trimeric adiponectin interact with T-cadherin in a Ca^2+^-dependent manner although T-cadherin interacts with the globular C1q domain of adiponectin also present in its trimeric form [6,108]. N-cadherin was identified as a CTRP9 receptor in the context of studies evaluating the combined cardioprotective activity of CTRP9 and adipose-derived stem cells (ADSCs, ref. [33]). These studies showed that CTRP9 promotes the proliferation, survival and migration of ADSCs empowering these cells to further boost the cardioprotective activity of CTRP9 after implantation [33]. Importantly, in vitro experiments suggested that the beneficial effects of CTRP9 on ADSCs are mediated via AMPK-independent ERK1/2 signaling and the upregulation of MMP9 and NRF2 [33]. Even more intriguingly, the beneficial CTRP9 signaling effects remained intact in AdipoR1 knockout ADSCs. Furthermore, cell-free coimmunoprecipitation experiments suggested that CTRP9 physically interact directly with N-cadherin and the inhibition of the CTRP9 N-cadherin interaction abrogated CTRP9-induced ERK signaling [33]. Furthermore, in macrophages, CTRP9 inhibits LPS-induced TLR4 signaling by preventing the formation of the LPS-induced LPS-TLR4-MD2-Myd88 signaling complex, an activity traced back to the competitive binding of LPS and CTRP9 to the TLR4-MD2 complex [32].

Adiponectin and some other CTRPs (CTRP1, CTRP5, CTRP13) were found to bind anionic phospholipids and sphingolipids such as phosphatidylserine, phosphatidylinositols, cardiolipin and ceramide-1-phophate suggesting that at least some CTRP family members function in lipid metabolism not only as adipokines but also as direct lipid binders [110].

CTRP3 was found to interact with the lysosomal-associated membrane protein 1 (LAMP1) in rat H4IIE hepatoma cells [18]. This interaction has also been confirmed in murine H9c2 cells where CTRP3 and LAMP1 promote proliferation and inhibit apoptosis in response to oxygen–glucose deprivation/reoxygenation treatment [111]. Moreover, there is evidence from inhibitor studies and coimmunoprecipitation experiments that CTRP3 and LAMP1 signal by JIP2/NUMB-assisted activation of cJun N-terminal kinases to protect from I/R injury [111,112]. Via its second C1q domain, which also mediates oligomerization of the molecule, CTRP4 interacts with nucleolin. The first C1q domain of CTRP4 is not involved in receptor binding and is mainly present in monomeric form and therefore not responsible for oligomerization of CTRP4 [23]. As indicated by its name, nucleolin is mainly found in the nucleolus; however, nucleolin was also found in other intracellular compartments and is secreted by yet poorly understood mechanisms to act as a cell-associated receptors for various viruses, some bacteria and some toxins [113]. Nucleolin is also exposed by apoptotic cells and there is evidence that it is recognized by CTRP4 to promote apoptotic cell clearance [23].

### 3.2. CTRP Regulated Signaling Pathways

The major signaling event triggered by adiponectin and most CTRPs is the activation of the evolutionarily conserved serine/threonine kinase adenosine monophosphate (AMP)-activated protein kinase (AMPK) (Table 4). AMPK consists of a heterotrimeric αβγ complex, with one of two different kinase domain-containing alpha subunits (α1, α2), one of two different beta subunits (β1, β2) which control cellular localization and complex assembly and one of three gamma subunits (γ1, γ2, γ3) harboring three functional and one non-functional cystathione β-synthetase AMP/ADP/ATP binding sites [114,115]. AMPK can be activated allosterically by AMP and/or by phosphorylation of the Thr172 in the alpha subunit, e.g., by liver kinase B1 (LKB1) or calmodulin-dependent protein kinase kinases (CaMKKs). AMPK senses the low ATP “energy state” of cells, which is associated for example with poor glucose supply or hypoxia, stimulates the activity of ATP-producing pathways and inhibits the activity of energy-consuming cellular programs. For example, at the molecular level, AMPK stimulates transcription and translocation of the insulin-stimulated glucose transporter GLUT4 and inhibits acetyl-CoA carboxylase (ACC), carnitine palmitoyl transferase I, TORC2, glycogen synthase, SREBP-1 and TSC2 thereby reducing the de novo synthesis of fatty acids, mitochondrial beta-oxidation and protein synthesis.

### 3.3. CTRP Signaling in the Heart

AMPK activation seems to be one of the central signaling events that mediates the cardioprotective effects of CTRPs [15,31,32,51,53,120]. CTRPs have been found to stimulate the activating Thr172 phosphorylation of AMPK in cardiomyocytes, thereby triggering Akt activation, translocation of GLUT1 and GLUT4 to the plasma membrane and the expression of glucose and fatty-acid metabolism-related enzymes (Figure 3) [15].

Knockdown experiments suggest, at least for CTRP7 and CTRP9, that AdipoR1 is the relevant receptor for these responses while AdipoR2 appears dispensable [15]. Stimulation of AMPK signaling by recombinant CTRPs could provide a therapeutic approach to restore glucose homeostasis and to reduce insulin resistance. In line with this, it has been observed that recombinant CTRP2, 7 and 9 stimulate AMPK-mediated inhibitory phosphorylation of Acetyl-CoA carboxylase (ACC) and thus impact fatty acid oxidation [15,31,51]. CTRP9 treatment of mice and cardiomyocytes after ischemia/reperfusion (I/R) was anti-apoptotic and interference with AMPK activation and AdipoR1 expression abolished this protective effect suggesting that CTRP9-induced AdipoR1/AMPK stimulation protects the heart from injury after I/R [31]. Similarly, in a lipopolysaccharide (LPS)-induced acute myocardial injury model, CTRP9 mediates anti-inflammatory protective effects, again via AdipoR1-mediated AMPK activation without involvement of AdipoR2 [51]. In cardiomyocytes and the H9c2 cell line, CTRP9 induces AMPK activation via AdipoR1 and AdipoR2 resulting in the upregulation of antioxidant enzymes and reduced ROS production in response to pro-hypertrophic triggers such as endothelin-1 or angiotensin II [53]. It is worth mentioning that adiponectin-induced AMPK activation and cardioprotection are reduced in mice lacking T-cadherin [121], opening the possibility that this adiponectin receptor is also involved in CTRP9 biology but this has not yet been evaluated. Similar to CTRP9, CTRP13 expression and exogenously added recombinant CTRP13 were shown to protect H9c2 cells from hypoxia/reoxygenation-induced oxidative stress through the AMPK-mediated activation of the transcription factor Nrf2 which promotes the transcription of antioxidant response element (ARE)-controlled detoxifying enzymes [120]. Furthermore, CTRP9 inhibits, at least partly via AMPK, Toll-like receptor 4 (TLR4) signaling and the Smad2,3-pathway in macrophages in the context of atrial fibrillation after MI [32,52]. Although CTRP9 largely elicits cardioprotective effects, it also promotes cardiac hypertrophy and left ventricular dilatation after constriction of the transverse aortic arch (see 1.). In this situation, CTRP9 engages ERK5 and its downstream target GATA4 to promote hypertrophy by yet unknown upstream mechanisms [45]. There is also in vitro evidence that CTRP9-induced AMPK-independent activation of PKA contributes to its antiapoptotic activity in the oxidative stress-induced cell death of H9c2 cells [54].

Besides activation of AMPK, a few other intracellular signaling pathways have been implicated in the CTRP-mediated protection of cardiomyocytes. CTRP3 protects from pressure overload-induced cardiac hypertrophy and heart dysfunction by the inhibition of the p38/CREB/ER stress pathway [62]. Furthermore, as already described in the receptor section, there is evidence that CTRP3 is cardioprotective through LAMP1-mediated activation of the JNK pathway [111,112]. CTRP15 shows anti-apoptotic and anti-inflammatory functions in the heart and in macrophages by AMPK-independent activation of the sphingosine-1phosphate (S1P)/cAMP/Akt signaling pathway [57]. Similarly, CTRP1 was shown to be antiapoptotic and anti-inflammatory in the heart after hypoxia/reoxygenation by AdipoR1-mediated activation of the S1P/cAMP pathway [59]. CTRPs do not only act on cardiomyocytes and macrophages in the heart. CTRP2 improves the outcome after I/R by increasing the tube formation and migration of endothelial cells by Akt activation and the upregulation of vascular endothelial growth factor receptor 2 [58]. In vitro experiments further suggest that CTRP15 reduces TGFβ1-mediated myofibroblast differentiation via the insulin receptor (IR)/IR substrate (IRS)-1/PI3K/Akt pathway [55]. Interestingly, the calreticulin/LPR1 co-receptor system contributes to protective Akt activation by adiponectin in cardiomyocytes in an AdipoR1/2-independent manner [122]. LPR1 has also been implicated in CTRP9-induced AMPK activation in H9c2 cells [123].

In sum, several CTRPs elicit cardioprotective effects in the heart, frequently via AdipoR1 or AdipoR2 but also by the help of receptors distinct from the AdipoRs. The cardioprotective activity of the CTRPs has been traced back to the activation of the AMPK signaling pathway but there is also clear evidence for the involvement of AMPK-independent signaling mechanisms.

### 3.4. CTRP Signaling in Adipose Tissue and in the Liver

Despite the large body of evidence for the crucial involvement of CTRPs in metabolic regulation obtained from CTRP knockout and transgenic mice or in vivo application of recombinant CTRPs, limited information is available for the direct signaling activities of CTRPs in adipose tissue or in the liver. Several CTRPs appear to directly improve insulin sensitivity, glucose and fatty acid utilization as well as endoplasmic reticulum (ER) stress and autophagy.

CTRP1 ameliorates adipose tissue insulin resistance by counteracting IRS-1 Ser1101 phosphorylation in mature primary human adipocytes. Ser1101 phosphorylation of IRS-1 results in the inhibition of insulin signaling and is most likely mediated by free fatty acid-activated PKCθ [124]. Subsequently, the glucose utilization rate of primary mature adipocytes improves upon cotreatment with insulin and CTRP1 [13]. In vivo, similar observations were made in transgenic CTRP1-overexpressing mice, which show improved insulin sensitivity [125], but also in low-fat diet-fed male CTRP1 knockout mice, which develop insulin resistance and show an impaired glucose metabolism [64]. However, S1101 phosphorylation of IRS-1 was not evaluated in these murine models. Instead, the molecular mechanisms by which CTRP1 affects insulin sensitivity were attributed to the AMPK signaling in skeletal muscle [64,125]. Similar to CTRP1, modest elevation of plasma CTRP3 levels by recombinant protein administration lowers glucose levels in normal and insulin-resistant ob/ob mice [126]. In this study, the normalization of glucose levels was ascribed to the activation of Akt and ERK1/2 signaling and the consecutive repression of hepatic gluconeogenic gene expression, while AMPK activity remained unchanged [126]. Akt and ERK1/2 are also activated by CTRP3 in mature 3T3-L1 adipocytes, resulting in a slight inhibitory effect on adipocyte differentiation [21]. In contrast to CTRP1 and CTRP3, CTRP5 appears to counteract insulin signaling in cultured adipocytes and myotubes by the attenuation of insulin-stimulated Akt phosphorylation [20]. Interestingly, while CTRP3 rather enhances the secretion of adiponectin from cultured adipocytes [117], CTRP5 inhibits adiponectin secretion [127], which indicates that CTRP3 and CTRP5 have counter-regulating activities in adipocytes. Consistent with its negative impact on glucose homeostasis, CTRP5 was identified as a novel negative regulator of white adipose tissue browning, most likely via the suppression of uncoupling protein 1 (UCP1) expression and adipocyte autophagy [80].

CTRP9, like CTRP1 and CTRP3, lowers blood glucose levels when overexpressed in obese mice and was shown to activate AMPK, Akt and ERK1/2 in cultured myotubes [50]. Lean mice are resistant to high-fat diet-induced weight gain upon the overexpression of CTRP9, display lower fasting insulin and glucose levels and an increased basal metabolism. Upon high-fat diet feeding insulin resistance and hepatic steatosis are prevented in CTRP9 transgenic mice [34]. In this study, CTRP9-induced AMPK activation enhanced oxidative phosphorylation in myotubes and reduced lipid accumulation in H4IIE hepatocytes [34]. Consequently, CTRP9 knockout mice display decreased insulin sensitivity and hepatic steatosis [128]. A more detailed mechanistic study utilizing primary human hepatocytes and hepatoma cell lines corroborated that CTRP9-mediated AMPK activation is important to induce autophagy, which is necessary to regulate basic metabolism in hepatocytes. Additionally, CTRP9 relieves fatty acid- or tunicamycin-induced ER stress [129]. While CTRP9 appears to alleviate ER stress, in vivo evidence from CTRP7 knockout mice indicates that this family member fosters oxidative and ER stress [26]. Indeed, recombinant CTRP7 increases free fatty acid-induced ROS levels and malondialdehyde levels in hepatoma cells. In addition, CTRP7 appears to antagonize insulin signaling since Tyr1131/1146 phosphorylation of the insulin receptor as well as Ser256 phosphorylation of Foxo1 were significantly reduced in insulin/CTRP7 co-treated hepatoma cells [130]. CTRP15 appears to promote fatty acid uptake into adipocytes and hepatocytes, partly by increasing the expression of genes that promote lipid uptake but the signaling pathways involved were not investigated in this study [43]. While CTRP9 has been shown to induce autophagic responses, CTRP15 suppresses starvation-induced autophagy in mouse liver and cultured hepatocytes via a PI3K/Akt/mTOR signaling [43].

Collectively, insulin signaling sensitizing (CTRP1, 3, 9) as well as antagonizing (CTRP5, 7) CTRP family members exist, which thereby affect glucose as well as lipid metabolism. Similar to the heart, the activation of AMPK plays an important role; however, for CTPR5, CTRP9 and CTRP15, evidence exist that this adipokine family has a major impact on the regulation of oxidative and ER stress as well as autophagy. Nonetheless, more mechanistic studies are needed to better understand the partly antagonistic activities.

## 4. Conclusions and Perspective

The physiology and pathophysiology of CTRPs have been intensively studied in animal models and argue for a crucial role of these ligands in autoimmune diseases, obesity, atherosclerosis and cardiac dysfunction. Indeed, the transgenic and viral expression of CTRPs as well as administration of recombinant CTRPs showed therapeutic activity in some disease models. Therefore, there is strong and broad preclinical evidence for considering CTRPs, CTRP receptors and CTRP-induced signaling pathways as promising therapeutic targets for a variety of pathologies. However, there are also considerable gaps in our understanding of CTRP signaling. There is only very limited knowledge about shared (and thus redundant) and specific (and thus unique) activities of CTRPs and CTRP receptors. Likewise, how CTRPs convert CTRP receptors from an inactive into an active state and the composition of CTRP receptor signaling complexes are largely unknown. This knowledge gap not only hinders the development of CTRP/CTRP receptor-targeting biologicals with high and specific activity but also the identification of low molecular weight drugs targeting CTRP-associated signaling pathways. The identification of yet unknown CTRP receptors, deeper insights in the relevance of the oligomeric state of CTRPs on the formation and activity of CTRP/CTRP receptor complexes and the elucidation of redundant and specific CTRP/CTRP receptor activities will help to better understand the pathophysiology of CTRPs empowering the rational development of therapeutic strategies targeting CTRPs, CTRP receptors or their signaling pathways.

## 5. Methods

As the basis for this narrative review, we carried out a search in PubMed using the following terms in title or abstract: “CTRP1 OR CTRP2 OR CTRP3 OR CTRP4 OR CTRP5 OR CTRP6 OR CTRP7 OR CTRP8 OR CTRP9 OR CTRP10 OR CTRP11 OR CTRP12 OR CTRP13 OR CTRP14 OR CTRP15 CTRP-1 OR CTRP-2 OR CTRP-3 OR CTRP-4 OR CTRP-5 OR CTRP-6 OR CTRP-7 OR CTRP-8 OR CTRP-9 OR CTRP-10 OR CTRP-11 OR CTRP-12 OR CTRP-13 OR CTRP-14 OR CTRP-15 OR ERFE OR erythroferrone OR Adipolin OR Cartonectin OR Cors26”. All articles from the beginning of the database up to December 2022 were considered; this way, approximately 860 publications were reached. Next, the retrieved publications were cross-searched with the term “receptor” to generate Table 1 and the term “mice” to establish Table 2. In addition, all receptors identified this way have been independently used as search terms to obtain further information about these molecules.

## Figures and Tables

**Figure 1 biomedicines-11-00559-f001:**
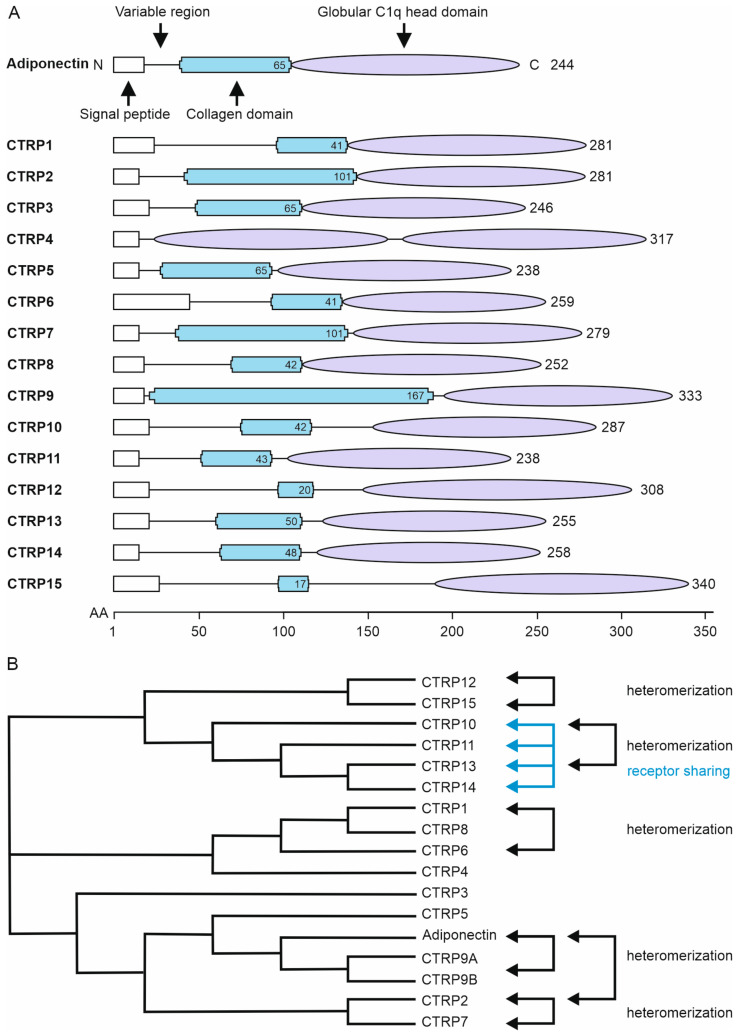
Domain architecture (**A**) and homology comparison (**B**) of CTRP family members. (**A**) CTRPs are characterized by a signal peptide (white) and a variable domain at the N terminus with conserved cysteine residues followed by a collagen domain (blue, the given number corresponds to the amino acid length of the collagen domain). As an exception, CTRP4 is lacking a collagen domain. At the C-terminus CTRP family members have a globular C1q head domain (purple) enabling trimerization. This domain is structurally related to the tumor necrosis factor (TNF) homology domain (THD) of ligands of the TNF superfamily (TNFSF) and the complement component C1q. Please note, in case of CTRP4 there are two C1q head domains. The number of amino acids of the CTRPs is indicated at the C terminus. (**B**) Cladogram of full-length CTRP family members. Strikingly, CTRP family members with a higher degree of sequence homology appear to form heteromers among each other. The same applies to CTRPs that share the same receptor. The cladogram was generated by multiple sequence alignment of protein sequences using ClustalW [94,95].

**Figure 2 biomedicines-11-00559-f002:**
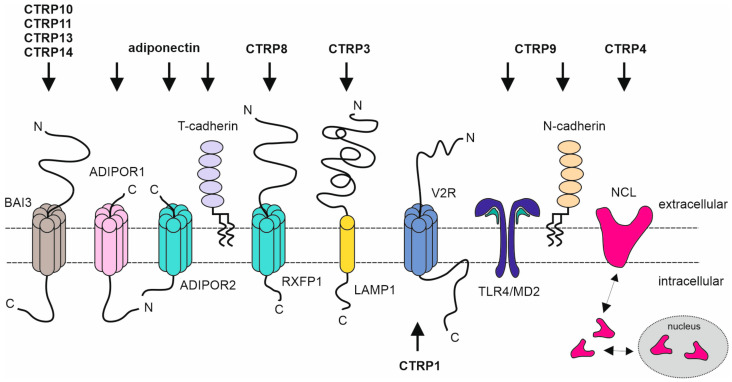
Schematic overview of known CTRP receptors. CTRP10, −11, −13 and −14 bind to and interact with the brain-specific angiogenesis inhibitor 3 (BAI3), a member of the cell-adhesion class of G protein-coupled receptors (GPCR). Adiponectin is a ligand for adiponectin receptors 1 and 2 (AdipoR1 and AdipoR2) and can also interact with T-cadherin. CTRP8 is described to interact with the GPCR relaxin/insulin-like family peptide receptor 1 (RXFP1). CTRP1 possibly binds intracellularly to the GPCR vasopressin receptor 2 (V2R) and lysosomal-associated membrane protein 1 (LAMP1) seems to be a receptor for CTRP3. There is evidence that CTRP9 binds to a complex consisting of Toll-like receptor 4 (TLR4) and myeloid differentiation protein 2 (MD2) and also interacts with N-cadherin. Moreover, it has been described that nucleolin (NCL) is a cell surface receptor for CTRP4.

**Figure 3 biomedicines-11-00559-f003:**
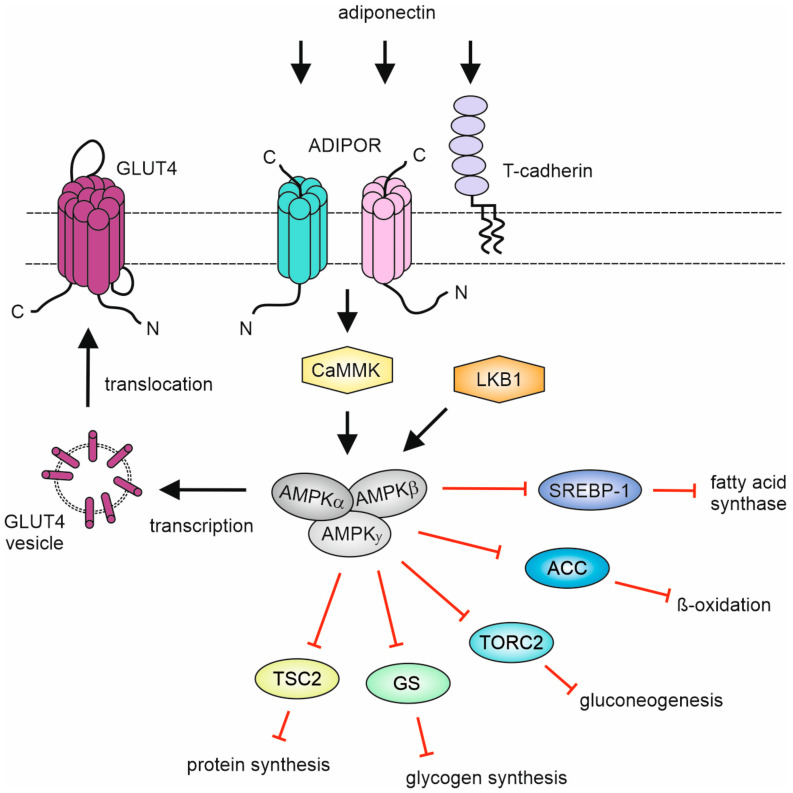
Schematic overview of CTRP-induced AMPK signaling. For details, see text.

**Table 1 biomedicines-11-00559-t001:** The CTRP family: producing and responding cell types.

CTRP	Receptor(s)	Producer Cells and Tissues	Responder Cells	Refs.
Adiponectin ADIPOQ Acrp30	AdipoR1 AdipoR2 T-cadherin	Adipocytes	Cardiomyocytes, skeletal muscle cells, β-cells, hepatocytes	[5,6,7,8,9,10]
CTRP1, C1QTNF1 GIP	V2R ? ^1^	Adipose tissue, placenta	Skeletal muscle cells, adipocytes	[11,12,13]
CTRP2, C1QTNF2		Adipose tissue, lung, liver, testis, uterus	Cardiomyocytes	[14,15]
CTRP3, C1QTNF3 Cors26 Cartducin Cartonectin	LAMP1	Adipocytes, monocytes	Endothelial cells, adipocytes	[16,17,18,19,20,21]
CTRP4, C1QTNF4	Nucleolin	Neurons	Monocytes, B-cells	[22,23]
CTRP5, C1QTNF5		Adipose tissue, eye: sub-retinal pigment epithelium (RPE)	Adipocytes	[11,20,24]
CTRP6, C1QTNF6		Placenta	Adipocytes	[25]
CTRP7, C1QTNF7	AdipoR1 ^2^	Liver	Cardiomyocytes	[15,26]
CTRP8, C1QTNF8	RXFP1	Lung, testis	Glioblastoma cells	[27,28,29]
CTRP9, C1QTNF9	AdipoR1 ^2^ TLR4/MD2 ^3^ N-cadherin	Adipocytes, heart	Cardiomyocytes, vascular endothelial cells, macrophages, skeletal muscle cells, hepatocytes	[15,30,31,32,33,34]
CTRP10, C1ql2	BAI3	Brain		[35,36]
CTRP11, C1ql4	BAI3	Adipose stroma, testis	Endothelial cells	[35,37,38]
CTRP12, C1QTNF12 FAM132A Adipolin		Adipose tissue	Hepatocytes	[39]
CTRP13, C1ql3	BAI3 NPTXR ^4^	Adipose tissue, brain	Pancreatic β-cells, cardiomyocytes, neurons	[15,35,40,41,42]
CTRP14, C1ql1	BAI3	Brain, testis	Endothelial cells	[35,36,37]
CTRP15, C1QTNF15 ERFE FAM132B Erythroferrone Myonectin		Skeletal muscle cells	Hepatocytes, macrophages, adipocytes	[43,44]

^1^ Evidence from pull-down experiments for binding of the C1q domain to the second intracellular loop of the V2 vasopressin receptor (V2R). ^2^ Indirect evidence from functional analysis of siRNA knockdown of receptor. ^3^ Indirect evidence from competition with LPS for binding to a TLR4-MD2 fusion protein. ^4^ CTRP13 indirectly binds to the neuronal pentraxin receptor via interaction with neuronal pentraxin 1. CTRP13 in turn recruits NPTX1 to BAI3.

**Table 4 biomedicines-11-00559-t004:** Recombinant (rec) CTRP-induced AMPK activation.

CTRP	AMPK Phosphorylation	AMPK Substrate/Downstream	Refs ^1^
CTRP1, tg, rec ^2^	Thr172	ACC (S79)	[116]
CTRP2, fl ^3^ rec	Thr172	ACC (S79), Akt(Thr308)	[15]
CTRP3	Thr172		[117]
CTRP5	Thr172	ACC (S79)	[118]
CTRP6	Thr172	ACC (S79)	[119]
CTRP7, fl rec	Thr172	ACC (S79), Akt(Thr308)	[15]
CTRP9, fl rec	Thr172	ACC (S79), Akt(Thr308)	[15]
CTRP13, fl rec	Thr172	Akt(Thr308)	[15]

^1^ References limited to first description(s) of AMPK activation. ^2^ rec, recombinant. ^3^ fl, full length.

## Data Availability

Not applicable.

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
