# Peer review of "Complement 1q/Tumor Necrosis Factor-Related Proteins (CTRPs): Structure, Receptors and Signaling"

_biomedicines, 2023, doi:10.3390/biomedicines11020559_

Round 1

Reviewer 1 Report

This is a review about the complement 1q (C1q) / tumor necrosis factor (TNF)-related protein family (CTRP). This family includes 16 members, being adiponectin the most representative and well-known one. The review is organized around 4  features, structure, possible receptors, mechanisms of cell signaling and functions/effects, although only 3 are mentioned at the title. Structure is well known, but the mechanisms involved in cell signaling is still partially unknown. Finally, the review focuses on the role of CTRP in heart and liver. I find the review useful updated and possible source of inspiration for people interesting in the field and in the possible therapeutical consequences of complete characterization of all CTRP receptors.

 I have only minor suggestions or questions for addressing before definitive acceptance.

 1)     It is early said that adiponectin stimulates the storage of fat in adipocytes and increases the insulin sensitivity of cells. According to that, the reduced production of adiponectin promotes the development of the metabolic syndrome (lines 36-38). In that way, it is suggested that disorders are due to low levels of adiponectin. Concerning to lipid metabolism, it is later said that the major signaling event triggered by adiponectin and most CTRPs is the activation of AMPK (lines 234-236). Storage of fat stimulation and activation of AMPK do not reconcile easily. This point should be clarified.

2)     On the other hand, is there any relation between CTRP and ACBP?? Bearing in mind the proposed role of this protein, this would be interesting.(see Acyl-CoA-binding protein (ACBP): the elusive 'hunger factor' linking autophagy to food intake. Cell Stress, 2019 Sep 24;3(10):312-318. doi: 10.15698/cst2019.10.200.

3)     The approximate length of the collagen domain should be given. As the structure of CTRP4 is different from the rest, the oligomerization of CTRP4 could be also different and therefore the interaction with nucleolin. These points would briefly discuss.

4)     The review has a large number of proteins termed as acronyms. Some of them are difficult to understand for general readers. A list of abbreviations would be welcome for a better comprehension.

5)     Tables (especially Table 2)  should be more compact to make them more friendly, if possiblewithin one page. (i.e.single spacing and smaller font sizes), 

Author Response

1) It is early said that adiponectin stimulates the storage of fat in adipocytes and increases the insulin sensitivity of cells. According to that, the reduced production of adiponectin promotes the development of the metabolic syndrome (lines 36-38). In that way, it is suggested that disorders are due to low levels of adiponectin. Concerning to lipid metabolism, it is later said that the major signaling event triggered by adiponectin and most CTRPs is the activation of AMPK (lines 234-236). Storage of fat stimulation and activation of AMPK do not reconcile easily. This point should be clarified.

We agree with the reviewer. Because the review is focusing on CTRP1-15 rather than on adiponectin, this phrasing was oversimplified and therefore misleading. We corrected this in the revised manuscript.

2)  On the other hand, is there any relation between CTRP and ACBP?? Bearing in mind the proposed role of this protein, this would be interesting.(see Acyl-CoA-binding protein (ACBP): the elusive 'hunger factor' linking autophagy to food intake. Cell Stress, 2019 Sep 24;3(10):312-318. doi: 10.15698/cst2019.10.200.

Thanks for your question. With respect to the involvement of CTRPs in the regulation of autophagy, there might be an interesting connection. However, since up to date no connection between CTRPs and ACBP/DBI has been proposed and would be highly speculative, we decided to not address this topic in our review.

3) The approximate length of the collagen domain should be given. As the structure of CTRP4 is different from the rest, the oligomerization of CTRP4 could be also different and therefore the interaction with nucleolin. These points would briefly discuss.

In Figure 1, approximate lengt of the collagen domain has been added with corresponding explanation in the figure legend (line 107/108). Discussion of CTRP4 structure, oligomerization and binding to nucleolin we inserted in lines 227-231. The first C1q domain of CTRP4 seems not to be involved in protein oligomerization or receptor binding.

4) The review has a large number of proteins termed as acronyms. Some of them are difficult to understand for general readers. A list of abbreviations would be welcome for a better comprehension.

Many thanks for your advice. We added a list of abbrevations as table 5 before the references (line 431).

5) Tables (especially Table 2)  should be more compact to make them more friendly, if possible) within one page. (i.e.single spacing and smaller font sizes), 

We agree with the reviewer and made all tables uniformly more compact (single spacing and larger width).

Reviewer 2 Report

Thank you very much for allowing me to review the review article entitled “CTRPs: structure, receptors and signaling” (biomedicines-2178191). This article is presented in the “Cell Biology and Pathology” Section of the Special Issue “10th Anniversary of Biomedicines-Recent Advances on Adipokines”.

The aim of this review is to review the biochemical properties of CTRPs and their receptors and the signaling pathways regulated by these molecules in the literature. With special attention, they address the problems for the therapeutic targeting of CTRP-related processes that arise from the fact that only a few CTRP receptors have been identified.

They found that the knowledge gap not only hinders the development of CTRP/CTRP receptor-targeting biologicals with high and specific activity but also the identification of low molecular weight drugs targeting CTRP-associated signaling pathways.

It is, therefore, a field of work in which we identify a great research dynamic, which is why a reflection (review) on the knowledge that is available at the present time is beneficial for planning the future of this research.

Comments:

I suggest that abbreviations not be used in the title for greater clarity of the content of the work for readers. I also suggest that information be indicated on what type of review has been carried out or what period it includes.

The summary must be structured stating the topic that the objective is, but also providing the methodology used and the main results and conclusions, so I suggest that it be modified.

In the introduction on lines 41 to 45, the references identified up to 2022 should be indicated more precisely, since making an open reference is a bit ambiguous.

Tables 1 and 2 are in the introduction when they are really the result of the review by the authors. I suggest that they be in the results section.

Please confirm that figures 1, 2 and 3 are made by the authors, if it still belongs to a revised work, please indicate where it was taken from and the authorization to use it.

Given that this is a review of an expanding field of work due to the importance that it implies, I suggest that you incorporate a methodology section that indicates the inclusion and exclusion criteria of the reviewed articles to facilitate future reviews on this.

Author Response

Comments:

I suggest that abbreviations not be used in the title for greater clarity of the content of the work for readers. I also suggest that information be indicated on what type of review has been carried out or what period it includes.

Thanks for your suggestions. We removed the abbrevations in the title. Moreover, we added an methodology section with information about the criteria of the publications considered and the included time span (line 411).

The summary must be structured stating the topic that the objective is, but also providing the methodology used and the main results and conclusions, so I suggest that it be modified.

We agree with the reviewer that the summary must be clearly structured. However, but since such a structure (methods, results, objectives, conclusion in a summery at the beginning) is not usual in this journal, we have decided to leave it at the given abstract.

In the introduction on lines 41 to 45, the references identified up to 2022 should be indicated more precisely, since making an open reference is a bit ambiguous.

Thank you for your suggestion and we agree with the reviewer that the open reference (Schäffler, A.; Buechler, C. CTRP family: linking immunity to metabolism. Trends Endocrinol Metab 2012, 23, 194-204, doi:10.1016/j.tem.2011.12.003.) is at the wrong place. For this reason we deleted it. Moreover all original references are listed in table 1. We decided to cite review articles (1-4) affecting diseases associated with CTRPs to avoid excessive referencing in the introduction, as this is not the main topic of this review.

Tables 1 and 2 are in the introduction when they are really the result of the review by the authors. I suggest that they be in the results section.

In order not to make the introduction too lengthy, we have decided to summarize the important background information in tables. Therefore and due to the fact that it is difficult to define a separate results section we think this is the right place for tables 1 and 2.

Please confirm that figures 1, 2 and 3 are made by the authors, if it still belongs to a revised work, please indicate where it was taken from and the authorization to use it.

You mentioned an important point. Of course we can confirm that the figures were created by ourselves. A corresponding note was added in line 422.

Given that this is a review of an expanding field of work due to the importance that it implies, I suggest that you incorporate a methodology section that indicates the inclusion and exclusion criteria of the reviewed articles to facilitate future reviews on this.

See first comment: we added an methodology section with information about the criteria of the publications considered and the included time span.

Round 2

Reviewer 2 Report

After carefully reading the new version of the manuscript entitled "CTRPs: structure, receptors and signaling" (biomedicines-2178191). This article is presented in the “Cell Biology and Pathology”, as well as the reviewers' response to the suggestion made.

I consider that it is an article that provides very interesting review information but that the authors have not yet located in time.

I would highlight the part of methodology that they have indicated, the work carried out is not a systematic review. a systematic review implies the use of more than one database as indicated by the PRISMA criteria. They also do not indicate the period they review since they cannot indicate until December 2022 without saying the temporary origin.

I also suggest that the summary be structured so that it can serve as an orientation for the readers.

Author Response

After carefully reading the new version of the manuscript entitled "CTRPs: structure, receptors and signaling" (biomedicines-2178191). This article is presented in the “Cell Biology and Pathology”, as well as the reviewers' response to the suggestion made. 

I consider that it is an article that provides very interesting review information but that the authors have not yet located in time.

I would highlight the part of methodology that they have indicated, the work carried out is not a systematic review. a systematic review implies the use of more than one database as indicated by the PRISMA criteria. They also do not indicate the period they review since they cannot indicate until December 2022 without saying the temporary origin.

The reviewer is right that we didn’t carry out a systematic review or meta-analysis. However, we want to stress and ask for understanding that our review is a narrative review which aims on giving a comprehensive overview concerning the state of current research on the structure, receptors and major signaling pathways of CTRPs. In so far, the selection of the articles considered in our review is eventually subjective and unsystematic.  Due to the broad and general nature of the topic of our review, it cannot be pressed in the clearly defined corset of a meta-analysis (comparing a defined intervention) or a systematic review evaluating a certain defined issue such as aetiology, prevalence, diagnosis or prognosis.

PRISMA „is an evidence-based minimum set of items for reporting in systematic reviews and meta-analyses. PRISMA primarily focuses on the reporting of reviews evaluating the effects of interventions, but can also be used as a basis for reporting systematic reviews with objectives other than evaluating interventions (e.g. evaluating aetiology, prevalence, diagnosis or prognosis).“ and not one to one applicable to narrative reviews as ours aiming on a rather broad scientific topic. 

Nevertheless, we have explained in the methodology section now in more detail how we basically retrieved the information / references considered in our review.  

I also suggest that the summary be structured so that it can serve as an orientation for the readers.

We re-structured the abstract according to the journal style into background - methods (type of review) - results – conclusion in a single paragraph w/o headings.

Round 3

Reviewer 2 Report

I have reviewed with interest the new version (v3) of the article entitled “CTRPs: structure, receptors and signaling” (biomedicines-2178191). This article is presented in the “Cell Biology and Pathology” section of the Special Issue “10th Anniversary of Biomedicines-Recent Advances on Adipokines”.

 Many thanks to the authors for their clarifications in the manuscript that facilitate the understanding of the work carried out.